# Analysis of Residents' Preparedness Protocols during Ebola Pandemic in Urban Environment

Emmanuel O. Amoo [1,*], Gbolahan A. Oni [1], Aize Obayan [2,3], Amos A. Alao [3], Olujide A. Adekeye [3], Gbemisola W. Samuel [1], Samuel A. Oyegbile [4] and Evaristus Adesina [5]

[1] Demography and Social Statistics, College of Management and Social Sciences, Covenant University, Ota 112233, Nigeria; gbolahan.oni@covenantuniversity.edu.ng (G.A.O.); gbemisola.samuel@covenantuniversity.edu.ng (G.W.S.)

[2] African Leadership Development Centre, Covenant University, Ota 112233, Nigeria

[3] Department of Psychology, College of Leadership and Development Studies, Covenant University, Ota 112233, Nigeria; amos.alao@covenantuniversity.edu.ng (A.A.A.); jide.adekeye@covenantuniversity.edu.ng (O.A.A.)

[4] Health Peak Medical Centre, Agege 100283, Nigeria; saoyegbile@gmail.com

[5] Department of Mass Communication, College of Management and Social Sciences, Covenant University, Ota 112233, Nigeria; evaristus.adesina@covenantuniversity.edu.ng

* Correspondence: emma.amoo@covenantuniversity.edu.ng

**Abstract:** Background: The study provided empirical analysis of the change in hygiene behavioural practices among community in Ogun and Lagos State with respect to Ebola outbreak in Nigeria. Methods: The data were extracted from a 2015 Cross-Sectional Survey on wellness, knowledge, attitude and practices towards the control and prevention of Ebola virus disease (EVD). Out of 1093 respondents selected in the main survey through simple random sampling technique within two enumeration areas (EAs), only 933 data cases were analyzable, leaving an attrition rate of 14.6%. The EAs represents the sampling points within the local government areas selected for the interviewed. Descriptive and inferential statistical techniques were both employed in the data analysis. Results: The results revealed high level of knowledge of EVD but over 70% were not aware of centre for treatment of EVD; 60.2% believed they cannot be susceptible to EVD. There were changes in certain practices that were canvassed and earlier adhered to during the outbreak. The practice of handshaking reduced, eating of hunted animals decreased only marginally by 6.9% and washing with soap increased by 4%. Conclusion: The study provides helpful insights for public health policy on possible mitigating strategies, especially in terms of behavioral risk factors that are prone to Ebola virus infections or other communicable diseases. The study emphasises that regular hand washing with soap and the use of sanitising agents including availability of treatment centres would be helpful in preventing the occurrence or re-occurrence of pandemic. The protocols identified in this study could be relevant to both medical personnel and the community for adoption especially as the unlikelihood of Ebola (or other pandemic) re-emergence have not been established.

**Keywords:** Ebola virus disease; knowledge; attitude; practices; wellness; men; sexual behaviour; environment



## 1. Introduction

Since its first incidence in 1976 at the northern part of Democratic Republic of the Congo in Africa, over 38 outbreaks of Ebola virus disease (EVD) have been recorded across the globe [1,2] The disease, between 2014 and 2016, hit adversely several countries in West African coast such as Guinea, Sierra Leone, Liberia and Nigeria and was noted as the first worst recorded outbreak of Ebola in sub-Saharan Africa [2,3]. Nigeria first record of the incidence of Ebola was announced on 20 July 2014 and by 20 October 2014, the World Health Organisation (WHO) declared Nigeria Ebola free [4,5]. While Nigeria experience was short lived, the intermittent emergence of the disease in other parts of

sub-Saharan Africa, in addition with the incursion of Lassa fever and COVID-19 have made EVD pandemic a serious threat in public health domain. Besides, the repeated occurrence of the Ebola disease around the world [1,2], in addition with the absence of any declaration that the disease cannot resurface again (wherever it has been once treated or warded off), has also created apprehension in public health domain. This has also make any analysis on the necessary protocols to prevent the re-occurrence or stop the spread of the disease a relevant and worthy scientific exercise.

Generally, infectious diseases caused more than 300 million illnesses and over five million deaths each year worldwide with communicable diseases causing more than 31% of deaths in developing countries [5–7]. The burden of communicable diseases (e.g., Ebola infection, Lassa fever and COVID-19) in developing nations was unacceptably high. While the proportional mortality in percentage of total deaths for all ages due to communicable diseases are 51%, 48%, 46% and 66% for Ghana, South Africa, Afghanistan and Nigeria respectively, the rates are as low as 5%, 4% and 6% in Canada, Italy and France, respectively [6,8]. In 2006 alone, almost 63% the population infected with HIV worldwide resided in sub-Saharan Africa in addition to the huge deaths caused by other communicable infections [8]. These deaths are largely preventable by adopting simple measures that could reduce risk factors for communicable diseases at community level. A report has indicated that the global Years of Life Lost (YLL) due to communicable diseases is as high as 69% [9] and more than 80% of this occurred in developing countries including Nigeria. The pre-Ebola era shows that Nigeria government expenditure on health ranges between 4.6% and 9.0% of her gross domestic product, while the private expenditure on health was as high as 60% and above [5,10–12]. Total per capita expenditure on health as at 2013 was estimated at $217 [13]. In addition to the domestic expenditure, the development assistance on health from 2000 and 2010 from donor countries is more than $27 billion [11,14,15]. These proportions must have heightened with the emergent of EVD as addition disease and without notable decline in the existing prevalent rates of other communicable diseases.

Total confirmed (including probable) cases of EVD were 14,124, 10,678, 3814 and 20 in Sierra Leone, Liberia, Guinea and Nigeria, respectively, but with the disease severity (fatality rate) as high as 28% (Sierra Leone), 45% (Liberia), 66.7% (Guinea) and 40% in Nigeria [1]. Reported cumulative cases of EVD in Guinea, Liberia, Sierra Leone, Nigeria and Mali totaled 28,645 out of which 11,324 resulted into death [1,13,16]. Ebola infection is feared among the citizenry especially due to the fact that it has no regional boundary and gender variation is a subject of controversy.

*Dynamic Pathways of Ebola Spread*

Ebola virus infection causes hemorrhagic fever in humans and primate with high mortality rate [17,18]. It is suspected that the viruses that cause EVD are peculiar to the region of sub-Saharan Africa region. The typical symptoms include but not limited to nausea, diarrhea, vomiting, damage to the liver [6,17,18]. It can lead to damaged vascular cells that form blood, skin tone, internal bleeding, body aches, fever and so on [6,17,18]. The virus that causes Ebola is transmitted by close and direct person-to-person contact with infected person or contact with infected person's bodily fluids such as blood, faeces or vomit or previously contaminated surfaces and objects including clothing, and unsanitised medical equipment that have been used to treat Ebola-infected patients [6,7,19,20]. Certain reports have indicated that Ebola kills relatively 20–90% of infected person [21] and such death usually occur between 6–16 days after the first symptoms appear [22]. It has also being confirmed that the virus has an incubation period of 2–21 days but could persist in semen for almost 70–90 days, thus the possibility of contracting the disease via sex [6,7,21]. Other mechanisms through which the virus could be transmitted includes contact with mouth or nose droplets from an infected individual [6,7,22].

Despite the numerous mechanisms of the spread of the disease with no consensus on one halting formula, the study proposes that sustenance of basic hygiene practices could prevent the spread of the virus [19]. Thus, the study investigated whether the community

sustained (still practicing) those hygienic protocols that were engaged in during the last Ebola outbreak as preparedness for preventing or halting the spread of pandemic disease.

The major control measures undertaken for EVD have been supportive care that range from rehydration and treatment of specific symptoms especially immune-based therapies and drug therapies. It also includes surveillance with urgent alert systems in cases of unexplained fever or deaths due to febrile illness [23,24]. The recent approval of vaccines and antibodies in DRC e.g., Inmazeb and Ebanga, Ervebo vaccine for adult ≥18 years and Zabdeno-and-Mvabea that are meant for individuals that are one year and older [7]. Ebola disease is characterized with high case-fatality rate, potential for repeated occurrence and rapid spread [21,25]. While timely surveillance and treatment are important control measures, continuous and regular analysis of hygiene protocols are crucial strategies for the prevention and control of diseases [24,26–28].

There have been numerous initiatives geared towards prevention of the spread of pandemic diseases, especially at it relates to Ebola in Nigeria and sub-Saharan Africa in general. Prominent are interventions by the West Africa Regional Centre for Surveillance and Disease Control (RCSDC) [29]. The organization is primarily concerned with prevention of diseases, especially in curbing the spread of EVD. It operated within the confine of West Africa Health Organisation (WAHO) and the Africa Centres for Disease Control [29]. In addition, there were also campaigns on hygiene practices by the Ministry of Health (MoH) and other organisations [30,31]. Despite all these efforts, EVD still infected large number of people that resulted in high mortality cases, including economic burden of relatively $53.19 billion occurring only in three countries (namely, Guinea, Liberia and Sierra Leone [32].

In reality, the success of most health programmes or campaigns especially in sub-Saharan Africa is contingent upon community disposition [31,33]. Besides, hygiene practices remains one of the uncontestable effective prevention methods against pandemic diseases, however, there are relatively little data on what behavioural changes take place in the presence of pandemic and what practices are sustained after such pandemic has been stopped or curtailed. Do people return to their former practices or sustain the mitigating or protective strategies adopted during such outbreak? The study specifically provided answers to certain boggling questions such as what are the levels of knowledge and practices in the selected study areas in respect of Ebola disease; what are the levels of community preparedness to prevent its spread? The understanding of the levels of knowledge and hygiene practices during Ebola disease outbreak is apt for development of policies and programs to prevent Ebola re-occurrence and emergence of any other epidemic-prone infectious diseases in Nigeria and by extension, other countries in sub-Sahara Africa.

## 2. Materials and Methods

### 2.1. Research Design

The data were extracted from a 2015 Cross-Sectional Survey on wellness, knowledge, attitude and practices towards the control and prevention of Ebola virus disease. The sample locations were drawn from two local government areas (LGA) randomly picked from Lagos and Ogun States (two states out of the 6 states in the South West geo-political zones of Nigeria). Each LGA is sub-divided into wards and the wards further divided into enumeration areas (EAs). The participants were drawn from the EAs. The EAs represents the sampling points within the local government areas selected for the study. Lagos is the second largest city in Africa, the most populous city in Nigeria. The city is the commercial nerve center of Nigeria and a major transit hub with air and sea ports of entry for Nigeria [34]. Ogun state is an adjoining state with also a major transit hub by road to other countries and linking other states from the southwest geopolitical zone of Nigeria [34]. The dense population and overburdened infrastructure in Lagos metropolis which has created inflow of population to adjoining communities that are located in Ogun state could easily produce environment where diseases can be easily transmitted and transmission sustained [30,31]. Overall, 1093 respondents were interviewed. The attrition

rate was 14.6% and only 933 data were analysed. Although, the best data referenced could be dataset from authorities such as MEASURE-DHS data, World Health Organisation data, UNICEF data, so on, but the use of different strategic assessment methodologies to capture indicators of vulnerability to dangerous epidemic-prone infectious diseases cannot be overemphasized [35]. It is also noted that considering the massive number of lives that the disease could affect, using diverse data collection approaches that follow basic research ethics should be encouraged. Thus, the dataset is regarded as apposite and timely as a strategic rapid response towards eradication of epidemic-prone infectious diseases through residents' information-based approach. In addition, the psychometric evaluation of questionnaire revealed satisfactory internal reliability with Cronbach's Alpha >0.6 for the components selected and the Test-retest reliability was found to be good, with all correlations range as $5.0 \pm 6.7$.

### 2.2. Participants and Data Analysis Procedure

In the survey, two local government areas each from the two states were randomly selected for the study. Each Local Government Area is sub-divided into wards and the ward further divided into enumeration areas. Two enumeration areas were purposively selected. As earlier indicated, an enumeration Area represents operational geographic areas for the collection, collation and dissemination of census data [36,37]. These enumeration areas are often used as national sampling frame for various surveys and censuses [37]. Respondents were selected using simple random sampling technique within two enumeration areas (EAs) that were randomly picked with each representing urban and rural areas.

Using the same approach, the households were chosen using the same random pattern and only one respondent in the family was interviewed. The research instruments and procedures were duly approved by the sponsoring institution ethical board—Covenant University Centre for Research, Innovation and Discovery (Ref: CUCRID-RG 008-11-14/FS 2014/2015).

### 2.3. Data Analysis Procedures

Data were analysed using both descriptive and inferential analytical techniques. Specifically, a three level of statistical analysis procedure was undertaken, namely univariate, bivariate and multivariate analysis. The univariate profiled the basic characteristics of the respondents and other individual variables of interest. The bivariate was employed to compute cross-tabulation between the identified hygiene behavioural practices during and after the outbreak. The logistic regression employed at the multivariate level, tested the model formulated on the responsiveness of susceptibility of respondents to EVD with respect to changes in certain hygienic practices.

## 3. Results

### 3.1. Background Information of the Respondents

Background variables selected include gender, age distribution, educational attainment, resident status, marital status and the usual place of residence. These were analysed according to the locations of study. The proportion of men is 44.6% against the women data of 55.4% (Table 1). The proportion is relatively the same between the two states. The rural/urban dichotomy reflected 47.6% (rural) and 52.4% (urban) for the total population but 51:49 and approximately 44 vs 56 in Lagos and Ogun state, respectively, as shown in Table 1. The unmarried in the two states were 31.9% (Lagos) and 36.4% (Ogun) which represents about 34% of the total percentage for the two states. Secondary education is the highest in terms of school enrolment of the communities. The proportion without formal education is lower (4.2%) in Lagos state compared with Ogun State (9.4%) as shown in Table 1. Overall, while individuals without formal education is 6.9%, primary education is 12.1, more than half of the respondents have attained secondary education (56.3%), while those with tertiary education are only 24.8% for the states.

**Table 1.** Demographic profile of respondents.

| Selected Demographic Variables | Lagos (*n* = 452) | | Ogun (*n* = 481) | | Total | |
|---|---|---|---|---|---|---|
| **Gender** | **Frequency** | **%** | **Frequency** | **%** | **Frequency** | **%** |
| Male | 204 | 45.1 | 212 | 44.1 | 416 | 44.6 |
| Female | 248 | 54.9 | 269 | 55.9 | 517 | 55.4 |
| **Marital status** | | | | | | |
| Married | 308 | 68.1 | 306 | 63.6 | 614 | 65.8 |
| Unmarried | 144 | 31.9 | 175 | 36.4 | 319 | 34.2 |
| **Place of residence** | | | | | | |
| Rural | 231 | 51.1 | 213 | 44.3 | 444 | 47.6 |
| Urban | 221 | 48.9 | 268 | 55.7 | 489 | 52.4 |
| **Education attainment** | | | | | | |
| No Education | 19 | 4.2 | 45 | 9.4 | 64 | 6.9 |
| Primary | 34 | 7.5 | 79 | 16.4 | 113 | 12.1 |
| Secondary | 280 | 61.9 | 245 | 50.9 | 525 | 56.3 |
| Tertiary | 119 | 26.3 | 112 | 23.3 | 231 | 24.8 |
| **Occupation distribution** | | | | | | |
| Trading | 182 | 40.3 | 224 | 46.6 | 406 | 43.5 |
| Skilled Artisan | 113 | 25.0 | 107 | 22.2 | 220 | 23.6 |
| Others (Civil servants, professionals, etc) | 157 | 34.7 | 150 | 31.2 | 307 | 32.9 |
| **Migration status** | | | | | | |
| In-Migrants | 22 | 4.9 | 39 | 8.1 | 61 | 6.5 |
| Natives | 430 | 95.1 | 442 | 91.9 | 872 | 93.5 |
| Total | 452 | 100.0 | 481 | 100.0 | 933 | 100.0 |

Source: Authors' fieldwork on community attitude, knowledge and practices. Survey on Ebola virus disease, 2015.

One out of every four respondents is trading in one form of business or the others. The proportion is 40.3% (Lagos state) and 46.6% (Ogun state). The proportion of individuals that are civil servants or professionals in Lagos (34.7%) is relatively higher than that of Ogun state (31.2%) as highlighted in Table 1. The question on the residency identified that 95.1% and 91.9% of the targeted population are residents of the study areas while only 4.9% and 8.1% in Lagos and Ogun states, respectively are persons who moved into the locations of study from another countries. They were capture if they have been in country for over 6 months.

### 3.2. Knowledge and Attitude on Ebola Virus Disease (EVD)

The respondents' knowledge in terms of the information they have on Ebola disease, including its mode of transmission, and the skills they have in the prevention of the spread or control of the disease were examined. Various indicators used include ever heard about EVD, respondent's level of information on EVD, belief that EVD is real, ability to recognize signs and symptoms of EVD, information about the EVD treatment centers, the specific hygiene practices, perception of respondents on: hand washing, handling or touching corpses and susceptibility to EVD.

The analysis revealed high level of awareness with 92.8% and 94% level of awareness in Lagos and Ogun state, respectively. Relatively the level of non-awareness is 1.2% higher in Lagos than Ogun state (Table 2). The rating of awareness level recorded higher level for Lagos (very much, 23.5%, and moderately as 36.3%) and individuals with little knowledge also constituted 36.3% as shown in Table 2. Similar rating was also recorded for Ogun state with 20.7%, 24.4% and, 49.8% for total respondents with very much knowledge, moderately and little knowledge on EVD, respectively (Table 2). The majority of the respondents believed that EVD existed and this is relatively the same in both states. However, the proportion could be said to be higher by 1.1% in Ogun state compared to the rate obtained

for Lagos state. Notwithstanding that the understanding of signs and symptoms of EVD is higher for both states, 11.1% (Ogun state) and 3.5% (Lagos state) could not be sure of understanding the signs and symptoms of EVD (Table 2). However, despite the high level of awareness and the belief of EVD existence, only 40.2% and 24.0% of the respondents are aware of any quarantine or treatment centre in Lagos and Ogun, respectively.

**Table 2.** Knowledge and Attitude on Ebola virus disease (EVD).

| Selected Indicators of Knowledge and Attitude | Lagos (*n* = 452) | | Ogun (*n* = 481) | |
|---|---|---|---|---|
| **Ever heard about Ebola virus disease** | **Number** | **%** | **Number** | **%** |
| Yes | 411 | 92.8 | 438 | 94.0 |
| No | 32 | 7.2 | 28 | 6.0 |
| **Total** | 443.0 | 100.0 | 466.0 | 100.0 |
| **Level of knowledge** | | | | |
| Very Much | 104 | 23.5 | 97 | 20.7 |
| Moderately | 161 | 36.3 | 114 | 24.4 |
| Very Little | 161 | 36.3 | 233 | 49.8 |
| Not at all | 17 | 3.8 | 24 | 5.1 |
| **Total** | 443 | 100.0 | 468 | 100.0 |
| **Believe that EVD existed** | | | | |
| Yes | 388 | 87.2 | 407 | 88.3 |
| No | 57 | 12.8 | 54 | 11.7 |
| **Total** | 445 | 100 | 461 | 100 |
| **Know the signs and symptoms of EVD** | | | | |
| Yes | 383 | 96.5 | 343 | 88.9 |
| No | 14 | 3.5 | 43 | 11.1 |
| **Total** | 397 | 100 | 386 | 100 |
| **Know EVD treatment centre** | | | | |
| Yes | 181 | 40.2 | 115 | 24.0 |
| No | 269 | 59.8 | 365 | 76.0 |
| **Total** | 450 | 100 | 480 | 100 |
| **Washed hands the last time you used toilet** | | | | |
| Not at all | 69 | 15.4 | 62 | 12.9 |
| With water only | 105 | 23.4 | 171 | 35.6 |
| With water and salt | 165 | 36.7 | 169 | 35.1 |
| With water and soap | 69 | 15.4 | 40 | 8.3 |
| With sanitizer | 41 | 9.1 | 37 | 7.7 |
| **Total** | 452 | 100 | 479 | 100.0 |
| **Who handles the corpse can contract EVD** | | | | |
| Yes | 303 | 70.3 | 322 | 72.2 |
| No | 128 | 29.7 | 124 | 27.8 |
| **Total** | 431 | 100 | 446 | 100 |
| **Think you are susceptible to contracting EVD** | | | | |
| Yes | 276 | 61.9 | 289 | 60.6 |
| No | 120 | 26.9 | 141 | 29.6 |
| **Total** | 446 | 100.0 | 477 | 100.0 |

Source: Authors' fieldwork on community attitude, knowledge and practices. Survey on Ebola virus disease, 2015.

Basic hygiene practices identified are hand washing but in various dimensions which include: washing hands with soap, washing hands with water only, washing hand with salt and washing hands with water and sanitiser. Proportion that are not practicing handwashing is higher in Lagos (15.4%) compared to 12.9% in Ogun state. Washing with

water only is higher in Ogun state (35.6%) compared to Lagos (23.4%) while hand washing with soap is higher in Lagos (15.4%) compared with 8.3%) in Ogun state.

The use of sanitiser is low and reduced by 1.2% for Ogun state than Lagos state. Further filtering questions used in the study showed that 29.7% of the respondents from Lagos disagreed with the question whether someone who handles or touches corpse can contract EVD. The frequency is 27.8% for Ogun State (see Table 2). The result also revealed that relatively, one in three respondents hold the view that handling of corpse is not related to possibility of being infected with EVD. The proportions that answer Yes or No to the question: Do you think you are susceptible to contract EVD are 61.9% and 60.6% in both Lagos and Ogun, respectively (Table 2).

### 3.3. Knowledge, Attitude and Practices in Both Pre- and Post-EVD Outbreak

The analysis on practices before and after Ebola outbreak revealed that some changes have taken place over few practices out of selected indicators investigated. Data were analysed without segregation by state. This is done to have holistic view of changes that have taken place in the entire region. The highlight of the result shows that the practice of handshaking reduced from 58.5% to 30.9% with a buffer on refusal to practices handshake which increase from 18.7% to 32.9% (Table 3). Also, while it can be reported that changed occurred in the practice of hand-washing, the practice increased from 9.4% to 20.9% among the respondents. However, in terms of hand-washing with soap, only 4% change was observed (Table 3). Specifically, from relatively 12.3% of the sample that used to wash hand with soap to only 16.3% as at the time of the survey.

**Table 3.** Practices before Ebola and After Ebola incidence.

| Selected Handwashing Practices | Before EVD Outbreak | | | After EVD Outbreak | | |
|---|---|---|---|---|---|---|
| | Always | Sometimes | Not at All | Always | Sometimes | Not at All |
| Practiced handshake/hugging | 58.5 | 22.9 | 18.7 | 30.9 | 36.1 | 32.9 |
| Washing hand with water | 9.4 | 19.2 | 71.4 | 15.5 | 30.5 | 54 |
| Washing hand with water and soap | 12.3 | 20.3 | 67.4 | 16.3 | 32.3 | 51.4 |
| Wash your hand with only water after using the toilet | 43.9 | 36.6 | 19.5 | 44.9 | 36.8 | 18.3 |
| Washing hand with water and soap after using the toilet | 54.2 | 38.3 | 7.5 | 59.9 | 35.7 | 4.4 |
| Washing hand with water only when I feel they are dirty | 32.2 | 48.6 | 19.3 | 34.1 | 46.4 | 19.6 |
| Washing hand with water and soap when I feel they are dirty | 41.4 | 46 | 12.6 | 46.4 | 43.1 | 10.5 |
| Use Sanitizer after hand shake/direct contact with people | 7.6 | 11.5 | 80.9 | 10.8 | 22.5 | 66.7 |
| Use hand sanitizer after using the toilet | 10.4 | 13.8 | 75.8 | 14.1 | 19.9 | 66 |
| Use hand sanitizer when I feel my hands are dirty | 6.9 | 11.7 | 81.3 | 10.8 | 20.5 | 68.7 |
| Handling or eating hunted wild animals | 26.1 | 31.2 | 42.7 | 19.2 | 27 | 53.8 |

Source: Authors' fieldwork on community attitude, knowledge and practices. Survey on Ebola virus disease, 2015.

In terms of the use of toilet and hand-washing, only a marginal 1% changed was recorded in washing hand with ordinary water after using toilet while the use of soap and water in washing hand increased from 54.2% to 59.9% (an increase change of 5.7%) as shown in Table 3. Regular washing hand with soap shows a change from 41.4% to 46.4% (5% change). Observation on the use of sanitizer revealed overall percentage change of only 3% (Table 3). Those who use hand sanitizer after handshake or people's contact increased

from 7.6% to 10.8%, individuals who use sanitizer after using toilet increased from 10.4% to 14.1%, the proportion who use sanitizer ordinarily to clean hands increased to 10.8% from 6.9% (Table 3). However, the proportion that eats hunted animals decreased marginally by 6.9% (from 26.1% to 19.1%) in the period before Ebola outbreak and post Ebola outbreak as shown in Table 3.

*3.4. Binary Logistic Regression Illustrating Respondent's Susceptibleness to EVD with Respect to Selected Practices*

At the multivariate level, the model formulated tested the responsiveness of susceptibility of respondents to EVD with respect to changes in certain hygienic practices. As earlier indicated, the dependent variable is the perception of self-susceptibility to EVD, captured as susceptibility = 1, no susceptibility = 0. The test technique employed is binary logistic and the independent variables are the identified different hygienic practices by the respondents. The results are presented in Table 4. The result revealed that those who practice handshaking or hugging sometimes or not at all are 0.566, and 0.624 (respectively) less likely to be susceptible to EVD compared to those who always observed the practice. The statistics specifically reflected odds ratio (OR) = 0.566, 95% CI (0.40–0.81), and OR = 0.624, 95% CI 0.43–0.90, respectively, and with correlation coefficients ($r$) = −0.569 and −0.472, respectively also. The practice of using water and soap to wash hand sometimes have negative likelihood of susceptibility to EVD. The correlation coefficient ($r$) shows −0.165, and odds ratio and confidence level (OR = 0.848, 95% CI 0.49–1.48). However, those who do not wash hand with water and soap are more 1.353 times more likely to be susceptible to EVD (OR = 1.353, 95% CI 0.84–2.18).

**Table 4.** Binary logistic regression illustrating respondent's susceptibleness to EVD with respect to selected practices.

| Selected Practices | B | Sig. | Exp(B) | 95% CI |
|---|---|---|---|---|
| **Practiced handshake/hugging** | | | | |
| Always (RC) | | | | |
| Sometimes | −0.569 | 0.002 | 0.566 | 0.40–0.81 |
| Not at all | −0.472 | 0.011 | 0.624 | 0.43–0.90 |
| **Wash hand with water and soap** | | | | |
| Always (RC) | | | | |
| Sometimes | −0.165 | 0.563 | 0.848 | 0.49–1.48 |
| Not at all | 0.302 | 0.214 | 1.353 | 0.84–2.18 |
| **Wash hand with water and soap after using the toilet** | | | | |
| Always (RC) | | | | |
| Sometimes | 0.022 | 0.955 | 0.979 | 0.47−2.06 |
| Not at all | 0.589 | 0.101 | 1.802 | 2.89–3.64 |
| **Washing hand with water** | | | | |
| Always (RC) | | | | |
| Sometimes | −1.065 | 0.000 | 0.345 | 0.20–0.58 |
| Not at all | 0.247 | 0.338 | 1.280 | 0.77–2.12 |
| **Use Sanitizer after contact with people** | | | | |
| Always (RC) | | | | |
| Sometimes | −0.151 | 0.558 | 0.860 | 0.52–1.42 |
| Not at all | 0.168 | 0.573 | 1.183 | 0.66–2.12 |

**Table 4.** *Cont.*

| Selected Practices | B | Sig. | Exp(B) | 95% CI |
|---|---|---|---|---|
| **Use hand sanitizer after using the toilet** | | | | |
| Always (RC) | | | | |
| Sometimes | −0.023 | 0.931 | 0.977 | 0.57–1.66 |
| Not at all | 0.080 | 0.779 | 1.083 | 0.62–1.89 |
| **Use hand sanitizer when I feel my hands are dirty** | | | | |
| Always (RC) | | | | |
| Sometimes | −0.050 | 0.853 | 0.951 | 0.56–1.61 |
| Not at all | 0.368 | 0.254 | 1.445 | 0.77–2.72 |
| **Handling/eating hunted wild animals** | | | | |
| Always (RC) | | | | |
| Sometimes | −0.235 | 0.427 | 0.791 | 0.44–1.41 |
| Not at all | −0.013 | 0.940 | 0.987 | 0.70–1.39 |
| Constant | 1.559 | 0.000 | 4.756 | |

−2 Log likelihood = 1162.417; Cox & Snell *R* Square = 0.091; Nagelkerke *R* Square = 0.123

Source: Authors' fieldwork on community attitude, knowledge and practices. Survey on Ebola virus disease, 2015.

Those who sometimes wash their hands with soap after using toilet are less likely to be susceptible to EVD with ($r$ = −0.022), while the odds ratio shows less likelihood (OR = 0.99, 95% CI 0.47–2.06) against those that avoid such practice with ($r$ = 0.589; OR = 1.80, CI 2.89–3.64) as shown in Table 4. Those that use sanitizer sometimes are less likely to be vulnerable to EVD (OR = 0.86, 95% CI 0.52–1.42), the correlation statistics shows ($r$ = −0.151). The individuals who do not use sanitizer are however 1.183 times more likely to be vulnerable to EVD (OR = 1.183, 95% CI 0.66–2.12), the correlation coefficient reveals $r$ = 0.168, implying a positive association between those would not use sanitizer and susceptibleness to epidemic. A similar test was also performed on the practice of using sanitizer whenever the hands are dirty. The analysis shows that this practice is negatively related ($r$ = −0.050) to susceptibleness to diseases. The supported evidences indicated (OR = 0.951, 95% CI 0.56–1.61). Those who do not observe this practice are however more likely to be exposed to epidemic disease (OR = 1.445, 95% CI 0.77–2.72), the correlation coefficient is positive ($r$ = 0.368). The result of the analysis on the handling or contact with bush meat or animals reflects a mix result. On one hand, there are negative correlations ($r$ = −0.235) with susceptibleness to disease (OR = 0.791, 95% CI 0.44–1.41), indicating less likelihood of being susceptible to epidemic. On the other hand, the analysis also brought out a negative correlation ($r$ = −0.13) and the odds ratio suggesting less probability of experiencing epidemic if such respondents are not in contact with animals (OR = 0.987, 95% CI 0.70–1.39).

## 4. Discussion

The study empirically highlighted the underlying dynamic mechanisms of the spread of Ebola virus disease and the preparedness of urban community towards controlling and curtailing outbreak of disease in the future. The findings are crucial for future mitigation efforts not only in urban area but also for the nation as a whole. The study emphasised that analysis of knowledge, attitude, especially the hygiene behavioural practices are crucial in the understanding of the spread of communicable diseases, including the management and prevention of their re-occurrences [29,31,38,39]. The study has therefore presented the general traditional quantitative information on community hygiene practices that may be prone to the spread of Ebola virus disease and by extension, other communicable diseases.

In this study, the preliminary literature showed that the outbreak of epidemic has been a consistent challenge in developing countries and has been ripping off the endowed human and natural material resources in terms of deaths, morbidity and health expen-

diture [7,8,13,23,40]. While Ebola prevalence has been currently halted [41] and suitable management procedures were initiated for COVID-19, guiding against their re-occurrence and spread through the monitoring of suspected indicators is important.

The role of residents in controlling or prevention of Ebola disease (and other diseases) seems indispensable. Individual resident should be responsible for their personal health and wellbeing and should play vital role in hygiene practices [42], hence the analysis of their perception is highly relevant to combating the disease. The high proportion of respondents (75%) who have no or lower level of education also suggests how far any intervention can succeed. Similar observations that relates to larger number of unemployed, full time housewives and unskilled artisans as found in this study could be integrated into future intervention initiatives. The most important way to reduce the spread of infections is hand washing. Washing hands with soap and running water is best practice because of the removal action of soap and water on transient microorganisms [4,7,19]. Also important is to get a vaccine for those infections and viruses that have one, when available and consistent environmental sanitation exercises [19,41]. However, only little changes were observed in the practice of hand-washing especially with soap in the communities studied. The usage of sanitiser is also not encouraging. While this could be due to low socio-economic status among the respondents such as low educational level and peasant occupation, proper understanding of the burden of infection might enhance a change in such attitude.

Among other salient findings from the study is the information on the migrants within the community. Thus, the authors ponder that if the disease was traced to trans-country movement and the random selection of resident' respondents produced immigrants as part of the respondents, examination of foreign nationals' living conditions, their healthcare access and the monitoring their in-and-out of country travelling is important for surveillance against a disease that has trans-national impacts. The point is more relevant when one considers unabated ravage of the diseases in neighbouring countries of Guinea, Liberia and Sierra Leone [13,42–44]. The knowledge of respondents on EVD symptoms and treatment centre indicated that such knowledge or information is not popular among the respondents. More than two-third of the total respondents (precisely 76%) in Ogun state have no idea about the Ebola treatment centres and relatively 60% in Lagos state. While this could be worrisome, it is a signal for the stakeholder for plausible intervention. It could also be assumed that probably, these category of respondents may not know that the disease requires specialized centres for the treatment of such infection. The observation also suggests that the facility may be completely unavailable. The knowledge on location of health facilities is vital when assessing the risk of a potential public health emergency [40,42]. It is thus essential that strategies to sensitise the community is important in knowing what do and what not to do for enduring prevention or control of diseases. Also, when considering the relatively little changes in certain practices, much might still be required in the area of public enlightenment concerning the disease.

## 5. Conclusion and Recommendations

The study provides helpful information for public health policy intervention especially in terms of hygiene behavioral risk factors that can make the community dwellers less susceptible to Ebola virus infections and other communicable diseases by extension. It specifically represents a baseline study that can guide in social mobilization strategies and activities towards reduction of the spread of EVD. Given the hundreds of travelers that travel to, from Nigeria to neighbouring countries through roads and paths, the crowded nature of various communities, unsanitised environment practices, sharing of toilets and open dump sites, appropriate guidance is necessary for effective intervention by governments and other stakeholders on how to reduce the risk of transmitting infections that are possibly spread through contact with body fluids or contaminated surfaces. The study emphasised regular hand washing with soap and the use of sanitising agents that could be helpful in preventing infections. It also highlighted that importance of availability of responsive treatment centers which must be known to the residents. The protocols identified in this study

could be relevant to both medical personnel and the community for adoption especially as the unlikelihood of pandemic re-emergence have not been established. Also, the report from this study would also be interesting and beneficial to scientific community especially on sustained handwashing-with-soap and not just ordinary handwashing-with-water. While affordable drugs and vaccines could be developed, readiness protocols through basic hygiene practices and health seeking attitude could reduce EVD spread and other diseases of its kind. It is also recommended that availability of treatment centres and vaccines are fundamental to effective response in curtailing health emergencies.

**Author Contributions:** Conceptualization, resources, A.O.; methodology, data management, formal analysis, and manuscript drafting, E.O.A.; validation, investigation, E.A.; G.W.S. and O.A.A.; funding acquisition, A.A.A. and G.A.O.; review and editing, O.A.A.; community reconnaissance, S.A.O. All authors have read and agreed to the published version of the manuscript.

**Funding:** The main research work received seed grants from Covenant University Centre for Research, Innovation and Discovery (CUCRID-RG 008-11-14/FS 2014/2015).

**Institutional Review Board Statement:** The Ethical Review Board of Covenant University Centre for Research, Innovation and Discovery (CUCRID) also reviewed the proposal and monitored the fieldwork activities.

**Informed Consent Statement:** Informed consent was obtained from all prospective respondents and where necessary, permission was also sought from husband or head of household before the consent form was assented.

**Data Availability Statement:** Data for this work is deposited with the funding centre: https://cucrid.covenantuniversity.edu.ng/.

**Acknowledgments:** The authors would like to appreciate Covenant University Centre for Research, Innovation and Discovery (CUCRID) for the funding of the project and the University Ethical/Research Committees for the screening, approval and the advice given at all stages of the survey. We would like to also pay tribute to late Aize Obayan (former Vice Chancellor, and former Director of African Leadership Development Centre) who supervised most of the fieldwork. Also, we recognize the participation and responses from our anonymous respondents in the various communities where the survey was conducted.

**Conflicts of Interest:** The authors declare no conflict of interest.

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
