# Peer review of "Analysis of Residents’ Preparedness Protocols during Ebola Pandemic in Urban Environment"

_sustainability, doi:10.3390/su13168934_

Round 1

Reviewer 1 Report

I have gone through the whole manuscript and overall, this is very clear, concise, and well-written. This article is presented for medical personnel and community to follow the present study rationale. I believe that this case report is interesting, and the topic is beneficial to the scientific community. I have only one suggestion that author should discuss more about the limitation and future implications of this study.

Author Response

Point-by-point Response to Reviewer’s Comments on manuscript entitled “Resident’s roles in the optimization of the preparedness protocols for Ebola pandemic in urban environment” (Manuscript ID - sustainability-1255851)

General:

  • We have attempted to remove the word optimisatn to make it general as thought by the reviewer.
  • Edited Title: Resident’s roles in preparedness protocols for pandemic in urban environment
  1. Abstract
  • The main observations have been made more conspicuous
  • The practices emphasised by our findings are those that could be crucial for curtailing pandemic spread.
  • The entire manuscript is an updated information on our chosen topic.

  1. Introduction

This section has been re-organized as suggested by the reviewer.

  • All highlighted and identified redundant statements have been completely removed, especially the sentence at line 114.
  • We quite agreed that there were missing words that distorted the readings of reader understanding of the flow of our thoughts. These have been completely filled out or re-phrased, or re-written accordingly.
  • Line 117 – Reference adequately supplied (Shuaib, Gunnala, 2014, Out, Ameh, 2018)
  • The introduction has generally been edited elaborately.
  • Few notes have been added to support objective of the study, especially as it related to the hygiene practices.

  1. Methods

The methods stated are the procedures we adopted in various states especially of the data collection procedures however, citations have been added as demanded by the reviewer.

  1. The data analysis procedures have been extended to multivariate level where one model was formulated to test the responsiveness of susceptibility of respondents to EVD with respect to changes in certain hygienic practices.

  1. Discussion has been beef up w
  2. Conclusion and recommendation were not affected as such.
  1. References

There is a few additional references.

Harkare, H. V., Corsi, D. J., Kim, R., Vollmer, S., & Subramanian, S. V. (2021). The impact of improved data quality on the prevalence estimates of anthropometric measures using DHS datasets in India. Scientific reports11(1), 1-13.

World Health Organization (2014). Nigeria is Now Free of Ebola Virus Transmission. Situation Assessment. World Health Organization; 2014. p. 20.

Centers for Disease Control and Prevention (CDC) 2019. National Center for Emerging and Zoonotic Infectious Diseases (NCEZID), Division of High-Consequence Pathogens and Pathology (DHCPP), Viral Special Pathogens Branch (VSPB) (2019).

Kamorudeen, R. T., & Adedokun, K. A. (2020). Ebola outbreak in West Africa, 2014 and 2016: Epidemic timeline, differential diagnoses, determining factors and lessons for future response. Journal of Infection and Public Health,  Volume 13, Issue 7, July 2020, pp956-962.

https://doi.org/10.1016/j.jiph.2020.03.014

Signed

Emmanuel O. Amoo

Reviewer 2 Report

Please see the detailed comments in attachment

Author Response

Point-by-point Response to Reviewer’s Comments on manuscript entitled “Resident’s roles in the optimization of the preparedness protocols for Ebola pandemic in urban environment” (Manuscript ID - sustainability-1255851)

General:

  • We have attempted to remove the word optimisation to make it general as thought by the reviewer.
  • Edited Title: Resident’s roles in preparedness protocols for pandemic in urban environment

  1. Abstract
  • The main observations have been made more conspicuous
  • The practices emphasised by our findings are those that could be crucial for curtailing pandemic spread.
  • The entire manuscript is an updated information on our chosen topic.

  1. Introduction

This section has been re-organized as suggested by the reviewer

  • All highlighted and identified redundant statements have been completely removed, especially the sentence at line 114.
  • We quite agreed that there were missing words that distorted the readings of reader understanding of the flow of our thoughts. These have been completely filled out or re-phrased, or re-written accordingly.
  • Line 117 – Reference adequately supplied (Shuaib, Gunnala, 2014, Out, Ameh, 2018)
  • The introduction has generally been edited elaborately.
  • Few notes have been added to support objective of the study, especially as it related to the hygiene practices.

  1. Methods

The methods stated are the procedures we adopted in various states especially of the data collection procedures however, citations have been added as demanded by the reviewer.

  1. The data analysis procedures have been extended to multivariate level where one model was formulated to test the responsiveness of susceptibility of respondents to EVD with respect to changes in certain hygienic practices.

  1. Discussion has been beef up w
  2. Conclusion and recommendation were not affected as such.

  1. References

There is a few additional references.

Harkare, H. V., Corsi, D. J., Kim, R., Vollmer, S., & Subramanian, S. V. (2021). The impact of improved data quality on the prevalence estimates of anthropometric measures using DHS datasets in India. Scientific reports11(1), 1-13.

World Health Organization (2014). Nigeria is Now Free of Ebola Virus Transmission. Situation Assessment. World Health Organization; 2014. p. 20.

Centers for Disease Control and Prevention (CDC) 2019. National Center for Emerging and Zoonotic Infectious Diseases (NCEZID), Division of High-Consequence Pathogens and Pathology (DHCPP), Viral Special Pathogens Branch (VSPB) (2019).

Kamorudeen, R. T., & Adedokun, K. A. (2020). Ebola outbreak in West Africa, 2014 and 2016: Epidemic timeline, differential diagnoses, determining factors and lessons for future response. Journal of Infection and Public Health,  Volume 13, Issue 7, July 2020, pp956-962.

https://doi.org/10.1016/j.jiph.2020.03.014

Signed

Emmanuel O. Amoo

Reviewer 3 Report

Article types: Article

Article title: Resident’s roles in the optimization of the preparedness protocols for Ebola pandemic in urban environment

Journal Name: Sustainability

The article titled: "Resident’s roles in the optimization of the preparedness protocols for Ebola pandemic in urban environment" explores the men’s role in the preparedness against the emerging pandemic of Ebola Virus Disease in Ogun State, Nigeria. It also studied the changes in men’s hygiene outbreak of Ebola Virus Disease. The review reports important information regarding the Ebola pandemic; however, the article should be improved to be accepted to be published. 

I have the following comments:

Major comments

  1. The title of the paper, in my opinion, could be more exact and informative. I suggest the authors reconsider and correct the title into a more generalized form. What the authors optimize? 
  2. The abstract is missing a few sentences indicating the main observations to date and the most important consequence of the research.
  3. The abstract did not centre on the protection measurements and also practices. The abstract should use the updated information regarding the topics.
  4. What do the two enumeration areas mean?
  5. The conclusion of the abstract is not clear. 
  6. The introduction section is vague and wordy. Many redundant sentences need to be deleted. 
  7. The introduction has to be re-organized and clearly illustrated to readers concerning contemporary protection measurements and challenges. In my opinion, authors should write a general introduction first (history, prevalence, etc.) and then write other details.
  8. The section of the introduction seems missing important information. This section needs deep modification with different medical hypotheses recommended different treatment approaches and gave convincing arguments for new discoveries urgently needed for treatment option Ebola.
  9. There are lots of redundant sentences repeating in the whole paper, making me feel the paper like a "cut and paste" from several other resources. For example, line 90: "Generally, infectious diseases caused more than"; line 84: "The history of Ebola in Nigeria started in July 2014 when a migrant (Liberian finance 84 ministry official Patrick Sawyer) died of the disease in Lagos few days after his arrival. 85 Few cases that were later discovered especially among the individuals who had contact 86 with the victim were pragmatically curtailed. Today, Nigeria has been given a clean slate 87 by the World Health Organization but there is no declaration by the body that the disease 88 cannot resurface if certain precautionary measures are not put in place". The sentences are not a way of scientific description. I suggest deleting sentences like these (there are more throughout the paper).
  10. The authors need to deploy the sentences more scientifically. This is a scientific research paper. For example, line 114: "There have numerous initiatives to curb and prevent the spread of Ebola in Nigeria and sub-Saharan Africa in general. The West Africa Regional Centre for Surveillance and Disease Control (RCSDC) that operates within the confine of West Africa Health Organisation (WAHO) and it is concerned with activities to curb EVD". I don't think predicting something without mentioning references is a good idea. Need to include references when authors made a statement. 
  11. Line 177: "There are also emerging 117 campaigns on hygiene practices from Ministry of Health (MoH) and organizations" no reference?
  12. The aim of this research needs to add a separate paragraph in the introduction section.
  13. "The history of Ebola in Nigeria started in July 2014 when a migrant (Liberian finance 84 ministry official Patrick Sawyer) died of the disease in Lagos few days after his arrival": In my opinion, from the previous section and this new section, there is a huge jump. In my eyes, it looks like two different papers. I think the authors should better connect the two parts of the manuscript. It is not clear why the authors want to discuss the general information without any references.
  14. I found no references in the methods section. The authors need to cite some references in this section.
  15. Also, the material and methods of writing need to extend on more analysis. This section is incomplete.
  16. Similar results section also limited. Only background information and Knowledge, Attitude and Practices is not enough. Need to add more analysis.
  17. The flow of the discussion is not perfect and unspecific. My advice is to make the sentences more lucid and legible for more productive comprehension. A rational discussion includes principal, relationship, and generalizations supported by the results.
  18. The conclusion section contains the details of the recommendation, not the summary information. Need to change the entire conclusion describing the summary of this research and the need for further in-depth perspectives (no references by their own words).
  19. I did not find any analysis or significance values for the validation in tables or figures.
  20. The number of references used is more minor.

Minor comments:

  1. A careful check of the English language by native expertise is required: there are several grammatical errors in the text.
  2. Many sentences are logically problematic. The authors should carefully check every sentence to ensure their sense, making sure no direct copy-paste.
  3. Spacing, punctuation marks, grammar, and spelling errors should be checked thoroughly in the manuscript. Typo occurs quite frequently throughout the manuscript.
  4. There is no reference in many cases, but since it's a research paper, the need to add relevant references.
  5. Novelty, in my opinion, is inadequate. The information is not new.
  6. Use correct and uniform terminology in the paper.
  7. Add a summary/aim of this research at the ending paragraph of the introduction section.
  8. Improper uses of abbreviations throughout the review.
  9. Line 255: The most important way to reduce the spread of infections is hand washing? Please make it clear.
  10. The list of references has to be designed and organized according to the journal guidelines.

Author Response

Point-by-point Response to Reviewer’s Comments on manuscript entitled “Resident’s roles in the optimization of the preparedness protocols for Ebola pandemic in urban environment”

General:

  • We have attempted to remove the word optimisation to make it general as thought by the reviewer.
  • Edited Title: Resident’s roles in preparedness protocols for pandemic in urban environment

  1. Abstract
  • The main observations have been made more conspicuous
  • The practices emphasised by our findings are those that could be crucial for curtailing pandemic spread.
  • The entire manuscript is an updated information on our chosen topic.

  1. Introduction

This section has been re-organized as suggested by the reviewer

  • All highlighted and identified redundant statements have been completely removed, especially the sentence at line 114.
  • We quite agreed that there were missing words that distorted the readings of reader understanding of the flow of our thoughts. These have been completely filled out or re-phrased, or re-written accordingly.
  • Line 117 – Reference adequately supplied (Shuaib, Gunnala, 2014, Out, Ameh, 2018)
  • The introduction has generally been edited elaborately.
  • Few notes have been added to support objective of the study, especially as it related to the hygiene practices.

  1. Methods

The methods stated are the procedures we adopted in various states especially of the data collection procedures however, citations have been added as demanded by the reviewer.

  1. The data analysis procedures have been extended to multivariate level where one model was formulated to test the responsiveness of susceptibility of respondents to EVD with respect to changes in certain hygienic practices.

  1. Discussion has been beef up w
  2. Conclusion and recommendation were not affected as such.

  1. References

There is a few additional references.

Harkare, H. V., Corsi, D. J., Kim, R., Vollmer, S., & Subramanian, S. V. (2021). The impact of improved data quality on the prevalence estimates of anthropometric measures using DHS datasets in India. Scientific reports11(1), 1-13.

World Health Organization (2014). Nigeria is Now Free of Ebola Virus Transmission. Situation Assessment. World Health Organization; 2014. p. 20.

Centers for Disease Control and Prevention (CDC) 2019. National Center for Emerging and Zoonotic Infectious Diseases (NCEZID), Division of High-Consequence Pathogens and Pathology (DHCPP), Viral Special Pathogens Branch (VSPB) (2019).

Kamorudeen, R. T., & Adedokun, K. A. (2020). Ebola outbreak in West Africa, 2014 and 2016: Epidemic timeline, differential diagnoses, determining factors and lessons for future response. Journal of Infection and Public Health,  Volume 13, Issue 7, July 2020, pp956-962.

https://doi.org/10.1016/j.jiph.2020.03.014

Signed

Emmanuel O. Amoo

Round 2

Reviewer 2 Report

The revised version has addressed some of my previous quesitons, however, it has ignored many important references in literature, as studying a hot topic.  And it is not suitable for pulication in the present form.

Author Response

Point-by-point Response to Reviewer’s Comment

General: It is pleasing to us that only one reviewer objected to most of our writings. The other expert accepted out manuscript. We appreciate this.

Title: The reviewer condemned the topic without any suggestion. That’s strange.

However, we have attempted to remove the word optimisation to make it general as thought by the reviewer.

Edited Title: Analysis of Residents’ preparedness protocols during Ebola pandemic in urban environment

Abstract

The main observations have been made more conspicuous.

The practices emphasised by our findings are those that could be crucial for curtailing pandemic spread.

The entire manuscript is an updated information on our chosen topic.

Introduction

This section has been re-organized and completely new as suggested by the reviewer

All highlighted and identified redundant statements have been completely removed, especially the sentence at line 114.

We quite agreed that there were missing words that distorted the readings of reader understanding of the flow of our thoughts. These have been completely filled out or re-phrased, or re-written accordingly.

Line 117 – Reference adequately supplied (Shuaib, Gunnala, 2014, Out, Ameh, 2018)

Few notes have been added to support objective of the study, especially as it related to the hygie practices.

The World Health Organisation estimated that over 20,000 human population could be infected and more than $490 million would be needed just for immediate response to the disease (Readon, 2014). The actual cost in Guinea, Liberia and Sierra Leone was estimated at $53.19billion (Huber, et al, 2018).

A new segment on data analysis procedures have been added.

binary logistic regression has been added to test respondent’s susceptiveness to EVD with respect to some practices.

Reviewer 3 Report

Article types: Article

Article title: Resident’s roles in the optimization of the preparedness protocols for Ebola pandemic in urban environment

Journal Name: Sustainability

The article titled: "Resident’s roles in the optimization of the preparedness protocols for Ebola pandemic in urban environment" explores the men’s role in the preparedness against emerging pandemic of Ebola Virus Disease in Ogun State, Nigeria. It also studied the changes in men’s hygiene outbreak of Ebola Virus Disease. The review reports important information regarding Ebola pandemic; however, the article should be improved to be accepted to be published. 

I have the following comments:

Major comments

  1. I did not find changes in the title. 
  2. Still no changes made in the abstract section. The abstract is missing a few sentences indicating the main observations to date and the most important consequence of the research.
  3. The abstract did not centre on the protection measurements and also practices. The abstract should use the updated information regarding the topics.
  4. What do the two enumeration areas mean? Need to answer.
  5. The conclusion of the abstract is not still clear. 
  6. The introduction section is unclear and wordy. Many redundant sentences need to be deleted. 
  7. The introduction has to be re-organized and clearly illustrated to readers concerning contemporary protection measurements and challenges.
  8. The section of the introduction seems missing important information. This section needs deep modification with different medical hypotheses recommended different treatment approaches and gave convincing arguments for new discoveries urgently needed for treatment option Ebola.
  9. The authors need to deploy the sentences more scientifically. This is a scientific research paper. For example, line 114: "There have numerous initiatives to curb and prevent the spread of Ebola in Nigeria and sub-Saharan Africa in general. The West Africa Regional Centre for Surveillance and Disease Control (RCSDC) that operates within the confine of West Africa Health Organisation (WAHO) and it is concerned with activities to curb EVD". I don't think predicting something without mentioning references is a good idea. Need to include references when authors made a statement. 
  10. The aim of this research needs to add a separate paragraph in the introduction section.
  11. The material and methods of writing need to extend on more analysis. This section is incomplete.
  12. The flow of the discussion is still not perfect and unspecific.
  13. The conclusion section contains the details of the recommendation, not the summary information. Need to change the entire conclusion describing the summary of this research and the need for further in-depth perspectives.

Minor comments:

  1. A careful check of the English language by native expertise is required: there are several grammatical errors in the text.
  2. Many sentences are logically problematic. The authors should carefully check every sentence to ensure their sense, making sure no direct copy-paste.
  3. Spacing, punctuation marks, grammar, and spelling errors should be checked thoroughly in the manuscript. Typo occurs quite frequently throughout the manuscript.
  4. Novelty, in my opinion, is inadequate. The information is not new.
  5. Use correct and uniform terminology in the paper.
  6. Add a summary/aim of this research at the ending paragraph of the introduction section.
  7. The list of references has to be designed and organized according to the journal guidelines.

Author Response

Point-by-point Response to Reviewer’s Comment

General: All comments have been attended to and the highlight are as indicated hereunder. However, detail of the correction are in the track-changes file and shown also in the Neat copy attached.

Title: We have attempted to remove the word optimisation to make it general as thought by the reviewer.

Edited Title: Analysis of Residents’ preparedness protocols during Ebola pandemic in urban environment

Abstract

The main observations have been made more conspicuous.

The practices emphasised by our findings are those that could be crucial for curtailing pandemic spread.

The entire manuscript is an updated information on our chosen topic.

Introduction

This section has been re-organized and completely new as suggested by the reviewer

All highlighted and identified redundant statements have been completely removed, especially the sentence at line 114.

We quite agreed that there were missing words that distorted the readings of reader understanding of the flow of our thoughts. These have been completely filled out or re-phrased, or re-written accordingly.

Line 117 – Reference adequately supplied (Shuaib, Gunnala, 2014, Out, Ameh, 2018)

Few notes have been added to support objective of the study, especially as it related to the hygie practices.

A segment on data analysis procedures have been added

Reasons for the choice of location have been provided

While Lagos is Africa's largest city, the commercial nerve center of Nigeria and a major transit hub with air and sea ports of entry for Nigeria, Ogun state is its adjoining states with also a major transit hub by road for the southwest geopolitical zone in Nigeria (Aliyu & Amadu, 2017). The dense population and overburdened infrastructure in Lagos metropolis and adjoining communities that are located in Ogun state is creating environment where diseases can be easily transmitted and transmission sustained (Shuaib, Gunnala, 2014, Out, Ameh, 2018).

Aliyu, A. A., & Amadu, L. (2017). Urbanization, cities, and health: the challenges to Nigeria–a review. Annals of African medicine16(4), 149. https://www.ncbi.nlm.nih.gov/pmc/articles/PMC5676403/

National Population Commission - NPC/Nigeria and ICF. (2019). 2018 Nigeria Demographic and Health Survey 2018. Abuja, Nigeria, and Rockville, Maryland, USA: NPC and ICF.

Omotayo, A. I., Ande, A. T., Oduola, A. O., Olakiigbe, A. K., Ghazali, A. K., Adeneye, A., & Awolola, S. T. (2021). Community Knowledge, Attitude and Practices on Malaria Vector Control Strategies in Lagos State, South-West Nigeria. Journal of Medical Entomology, 58(3), 1280-1286.

Seale, H., Heywood, A. E., Leask, J., Sheel, M., Thomas, S., Durrheim, D. N., ... & Kaur, R. (2020). COVID-19 is rapidly changing: Examining public perceptions and behaviors in response to this evolving pandemic. PloS one, 15(6), e0235112.

Yanti, B., Mulyadi, E., Wahiduddin, W., Novika, R. G. H., Arina, Y. M. D. A., Martani, N. S., & Nawan, N. (2020). Community knowledge, attitudes, and behavior towards social distancing policy as prevention transmission of COVID-19 in indonesia. Jurnal Administrasi Kesehatan Indonesia, 8, 4-14.

Round 3

Reviewer 2 Report

The authors have empirically studied the Ebola pandemic,however,  the lack of in-depth understanding of the underlying mechanism of Ebola  spreading  dynamics, as well as the  effects of the prevention measures hinders this work being a hihg-impact research.  I suggest such work should be mentioned and additionally applied to this paper. 

Author Response

Point-by-Point Response to Reviewers’ Comments

Comments from Reviewer 2

The authors have empirically studied the Ebola pandemic however, the lack of in-depth understanding of the underlying mechanism of Ebola spreading dynamics, as well as the effects of the prevention measures hinders this work being a high-impact research.  I suggest such work should be mentioned and additionally applied to this paper. 

Response (to Reviewer No 2)

These point have been assembled together and designated as a sub-heading tagged “Dynamic pathways of Ebola spread” and the section contain the following:

Dynamic pathways of Ebola spread

Ebola virus infection causes hemorrhagic fever in humans and primate with high mortality rate (Lawrence et al., 2017; Oestereich et al., 2014). It is suspected that the viruses that cause EVD are peculiar to the region of sub-Saharan Africa region. The typical symptoms include but not limited to nausea, diarrhea, vomiting, damage to the liver, can lead to damaged vascular cells that form blood, skin tone, internal bleeding, body aches, fever and so on (Lawrence et al., 2017; Oestereich et al., 2014; World Health Organisation, 2014). The disease of Ebola is caused by virus that is transmitted by close and direct person-to-person o infected bodily fluid of infected person.  It is also spread though contact with infected person’s blood, faeces or vomit or previously contaminated surfaces and objects including clothing, and unsanitised medical equipment that have been used to treat Ebola-infected patients (Li, 2017; World Health Organisation, 2014, 2021). It has also being confirmed that the virus has an incubation period of 2-21 days but could persist in semen for almost 70-90 days, the possibility of contracting the disease via sex is very high (World Health Organisation, 2014, 2021). Other mechanisms through which the virus could be transmitted includes contact with mouth or nose droplets from an infected individual (World Health Organisation, 2014, 2021).

Despite the numerous mechanisms of the spread of the disease with no consensus on one halting formula, the study proposes that sustained basic hygiene practices could prevent the spread of the virus. Thus, study investigated whether the community sustained (are still practicing) those hygienic practices that were engaged in during the last Ebola outbreak as preparedness for preventing or halting the spread of pandemic disease.

References

References format of the journal has been followed.

In addition, a few literature has been added to the study. These include the following:

Isibor I, Bassey B. Seminal Fluid: Potential Sources of Ebola Virus Disease Transmission in the Population. Pediatr Infect Dis Open Access. 2016;1(4):27.

Lawrence, P., Danet, N., Reynard, O., Volchkova, V., & Volchkov, V. (2017). Human transmission of Ebola virus. Current opinion in virology, 22, 51-58.

Li, L. (2017). Transmission dynamics of Ebola virus disease with human mobility in Sierra Leone. Chaos, Solitons & Fractals, 104, 575-579. https://doi.org/10.1016/j.chaos.2017.09.022

Oestereich, L., Lüdtke, A., Wurr, S., Rieger, T., Muñoz-Fontela, C., & Günther, S. (2014). Successful treatment of advanced Ebola virus infection with T-705 (favipiravir) in a small animal model. Antiviral research, 105, 17-21.

World Health Organisation (2014). What we know about transmission of the Ebola virus among humans. Ebola situation assessment. October 6, 2014. https://www.who.int/news/item/06-10-2014-what-we-know-about-transmission-of-the-ebola-virus-among-humans

Reviewer 3 Report

Article types: Article

Article title: Resident’s roles in the optimization of the preparedness protocols for Ebola pandemic in urban environment

Journal Name: Sustainability

The article titled: "Resident’s roles in the optimization of the preparedness protocols for Ebola pandemic in urban environment" explores the men’s role in the preparedness against the emerging pandemic of Ebola Virus Disease in Ogun State, Nigeria. It also studied the changes in men’s hygiene outbreak of Ebola Virus Disease. The review reports important information regarding the Ebola pandemic; however, the article should be improved to be accepted to be published. 

The authors addressed most of my comments. I have the following minor comments:

Minor comments

  1. The authors need to deploy the sentences more scientifically. This is a scientific research paper. For example, line 195: "The analysis revealed overwhelming level of awareness…". Please change.
  2. The list of references has to be designed and organized according to the journal guidelines.
  3. Spacing, punctuation marks, grammar, and spelling errors should be checked thoroughly in the manuscript. Typo occurs quite frequently throughout the manuscript.
  4. Evaristus Adesina author has no affiliation??

Author Response

Comments from Reviewer 3

The article titled: "Resident’s roles in the optimization of the preparedness protocols for Ebola pandemic in urban environment" explores the men’s role in the preparedness against the emerging pandemic of Ebola Virus Disease in Ogun State, Nigeria. It also studied the changes in men’s hygiene outbreak of Ebola Virus Disease. The review reports important information regarding the Ebola pandemic; however, the article should be improved to be accepted to be published. 

The authors addressed most of my comments. I have the following minor comments:

Minor comments

S/N

Comments from Reviewer 3

Authors’ responses

1

The authors need to deploy the sentences more scientifically. This is a scientific research paper. For example, line 195: "The analysis revealed overwhelming level of awareness…". Please change.

This has been changed

2

The list of references has to be designed and organized according to the journal guidelines

Zotero Reference manager has been used for both the citations and the references iin compliance with the reference format.

3

Spacing, punctuation marks, grammar, and spelling errors should be checked thoroughly in the manuscript. Typo occurs quite frequently throughout the manuscript.

The entire manuscript has been thoroughly edited now

4

Evaristus Adesina author has no affiliation

Evaristus Adesina’s affiliation has been added.

.

References

References format of the journal has been followed.

In addition, a few literature has been added to the study. These include the following:

Isibor I, Bassey B. Seminal Fluid: Potential Sources of Ebola Virus Disease Transmission in the Population. Pediatr Infect Dis Open Access. 2016;1(4):27.

Lawrence, P., Danet, N., Reynard, O., Volchkova, V., & Volchkov, V. (2017). Human transmission of Ebola virus. Current opinion in virology, 22, 51-58.

Li, L. (2017). Transmission dynamics of Ebola virus disease with human mobility in Sierra Leone. Chaos, Solitons & Fractals, 104, 575-579. https://doi.org/10.1016/j.chaos.2017.09.022

Oestereich, L., Lüdtke, A., Wurr, S., Rieger, T., Muñoz-Fontela, C., & Günther, S. (2014). Successful treatment of advanced Ebola virus infection with T-705 (favipiravir) in a small animal model. Antiviral research, 105, 17-21.

World Health Organisation (2014). What we know about transmission of the Ebola virus among humans. Ebola situation assessment. October 6, 2014. https://www.who.int/news/item/06-10-2014-what-we-know-about-transmission-of-the-ebola-virus-among-humans
